# Social anxiety in young people: A prevalence study in seven countries

**Philip Jefferies** *, **Michael Ungar**

Resilience Research Centre, Faculty of Health, Dalhousie University, Halifax, Nova Scotia, Canada

* Philip.jefferies@dal.ca

## Abstract

Social anxiety is a fast-growing phenomenon which is thought to disproportionately affect young people. In this study, we explore the prevalence of social anxiety around the world using a self-report survey of 6,825 individuals (male = 3,342, female = 3,428, other = 55), aged 16–29 years (M = 22.84, SD = 3.97), from seven countries selected for their cultural and economic diversity: Brazil, China, Indonesia, Russia, Thailand, US, and Vietnam. The respondents completed the Social Interaction Anxiety Scale (SIAS). The global prevalence of social anxiety was found to be significantly higher than previously reported, with more than 1 in 3 (36%) respondents meeting the threshold criteria for having Social Anxiety Disorder (SAD). Prevalence and severity of social anxiety symptoms did not differ between sexes but varied as a function of age, country, work status, level of education, and whether an individual lived in an urban or rural location. Additionally, 1 in 6 (18%) perceived themselves as not having social anxiety, yet still met or exceeded the threshold for SAD. The data indicate that social anxiety is a concern for young adults around the world, many of whom do not recognise the difficulties they may experience. A large number of young people may be experiencing substantial disruptions in functioning and well-being which may be ameliorable with appropriate education and intervention.

**Data Availability Statement:** All data files are available from the Open Science Framework repository (DOI: 10.17605/OSF.IO/VCNF7).

**Funding:** The author(s) received no specific funding for this work.

## Introduction

Social anxiety occurs when individuals fear social situations in which they anticipate negative evaluations by others or perceive that their presence will make others feel uncomfortable [1]. From an evolutionary perspective, at appropriate levels social anxiety is adaptive, prompting greater attention to our presentation and reflection on our behaviours. This sensitivity ensures we adjust to those around us to maintain or improve social desirability and avoid ostracism [2]. However, when out of proportion to threats posed by a normative social situation (e.g., interactions with a peer group at school or in the workplace) and when impairing functioning to a significant degree, it may be classified as a disorder (SAD; formerly 'social phobia'; [3]). The hallmark of social anxiety in western contexts is an extreme and persistent fear of embarrassment and humiliation [1, 4, 5]. Elsewhere, notably in Asian cultures, social anxiety may also manifest as embarrassment of others, such as *Taijin kyofusho* in Japan and Korea [6]. Common concerns involved in social anxiety include fears of shaking, blushing, sweating,

**Competing interests:** I have read the journal's policy and the authors of this manuscript have the following competing interests: Unilever funds the lead author's research fellowship at Dalhousie University's Resilience Research Centre, though in no way have they directed this research, its analysis or the reporting or results.

appearing anxious, boring, or incompetent [7]. Individuals experiencing social anxiety visibly struggle with social situations. They show fewer facial expressions, avert their gaze more often, and express greater difficulty initiating and maintaining conversations, compared to individuals without social anxiety [8]. Recognising difficulties can lead to dread of everyday activities such as meeting new people or speaking on the phone. In turn, this can lead to individuals reducing their interactions or shying away from engaging with others altogether.

The impact of social anxiety is widespread, affecting functioning in various domains of life and lowering general mood and wellbeing [9]. For instance, individuals experiencing social anxiety are more likely to be victims of bullying [10, 11] and are at greater risk of leaving school early and with poorer qualifications [11, 12]. They also tend to have fewer friends [13], are less likely to marry, more likely to divorce, and less likely to have children [14]. In the workplace, they report more days absent from work and poorer performance [15].

A lifetime prevalence of SAD of up to 12% has been reported in the US [16], and 12-month prevalence rates of .8% have been reported across Europe [17] and .2% in China [18]. However, there is an increasing trend to consider a spectrum of social anxiety which takes account of those experiencing subthreshold or subclinical social anxiety, as those experiencing more moderate levels of social anxiety also experience significant impairment across different domains of functioning [19–21]. Therefore, the proportion of individuals significantly affected by social anxiety, which include a substantial proportion of individuals with undiagnosed SAD [8], may be higher than current estimates suggest.

Studies also indicate younger individuals are disproportionately affected by social anxiety, with prevalence rates at around 10% by the end of adolescence [22–24], with 90% of cases occurring by age 23 [16]. Higher rates of social anxiety have also been observed in females and are associated with being unemployed [25, 26], having lower educational status [27], and living in rural areas [28, 29]. Leigh and Clark [30] have explored the higher incidence of social anxiety in younger individuals, suggesting that moving from a reliance on the family unit to peer interactions and the development of neurocognitive abilities including public self-consciousness may present a period of greater vulnerability to social anxiety. While most going through this developmentally sensitive period are expected to experience a brief increase in social fears [31], Leigh and Clark suggest that some who may be more behaviourally inhibited by temperament are at greater risk of developing and maintaining social anxiety.

Recent accounts suggest that levels of social anxiety may be rising. Studies have indicated that greater social media usage, increased digital connectivity and visibility, and more options for non-face-to-face communication are associated with higher levels of social anxiety [32–35]. The mechanism underpinning these associations remains unclear, though studies have suggested individuals with social anxiety favour the relative 'safety' of online interactions [32, 36]. However, some have suggested that such distanced interactions such as via social media may displace some face to face relationships, as individuals experience greater control and enjoyment online, in turn disrupting social cohesion and leading to social isolation [37, 38]. For young people, at a time when the development of social relations is critical, the perceived safety of social interactions that take place at a distance may lead some to a spiral of withdrawal, where the prospect of normal social interactions becomes ever more challenging.

Therefore, in this study, we sought to determine the current prevalence of social anxiety in young people from different countries around the world, in order to clarify whether rates of social anxiety are increasing. Specifically, we used self-report measures (rather than medical records) to discover both the frequency of the disorder, severity of symptoms, and to examine whether differences exist between sexes and other demographic factors associated with differences in social anxiety.

## Materials and methods

### Design

This study is a secondary analysis of a dataset that was created by Edelman Intelligence for a market research campaign exploring lifestyles and the use of hair care products that was commissioned by Clear and Unilever. The original project to collect the data took place in November 2019, where participants were invited to complete a 20-minute online questionnaire containing measures of social anxiety, resilience, social media usage, and questions related to functioning across various life domains. Participants were randomly recruited through the market research companies Dynata, Online Market Intelligence (OMI), and GMO Research, who hold nationally representative research panels. All three companies are affiliated with market research bodies that set standards for ethical practice. Dynata adheres to the Market Research Society code of conduct; OMI and GMO adhere to the ESOMAR market research code of conduct. The secondary analyses of the dataset were approved by Dalhousie University's Research Ethics Board.

### Participants

There were 6,825 participants involved in the study (male = 3,342, female = 3,428, other = 55), aged 16–29 years (M = 22.84, SD = 3.97), from seven countries selected for their social and economic diversity (Brazil, China, Indonesia, Russia, Thailand, US, and Vietnam) (see Table 1 for full sample characteristics). Participant ages were collected in years, but some individuals aged 16–17 were recruited through their parents and their exact age was not given. They were assigned an age of 16.5 years in order to derive the mean age and standard deviation for the full sample.

### Procedure

Email invitations to participate were sent to 23,346 young people aged 16–29, of whom 76% (n = 17,817) were recruited to take the survey. These were panel members who had previously registered and given their consent to participate in surveys. Sixty-five percent of respondents were ineligible, with 10,816 excluded because they or their close friends worked in advertising, market research, public relations, journalism or the media, or for a manufacturer or retailer of haircare products. A further 176 respondents were excluded for straight-lining (selecting the same response to every item of the social anxiety measure, indicating they were not properly engaged with the survey; [39]). The final sample comprised 6,825 participants and matched

**Table 1. Sample characteristics.**

|  | Male | Female | Other [a] | Total |
|---|---|---|---|---|
| Brazil | 479 | 491 | 7 | 977 |
| China | 486 | 500 | 6 | 992 |
| Indonesia | 494 | 457 | 8 | 959 |
| Russia | 475 | 500 | 8 | 983 |
| Thailand | 469 | 487 | 12 | 968 |
| US | 452 | 500 | 10 | 962 |
| Vietnam | 487 | 493 | 4 | 984 |
| *Total* | *3,342* | *3,428* | *55* | *6,825* |

[a] "Other" includes individuals who selected non-binary (n = 17), prefer to self-describe (n = 7), and prefer not to say (n = 31).

quotas for sex, region, and age, to achieve a sample with demographics representative of each country.

Participants were compensated for their time using a points-based incentive system, where points earned at the end of the survey could be redeemed for gift cards, vouchers, donations to charities, and other products or services.

## Measures

The survey included the 20-item self-report Social Interaction Anxiety Scale (SIAS; [40]). Based on the DSM, the SIAS was originally developed in conjunction with the Social Phobia Scale to determine individuals' levels of social anxiety and how those with SAD respond to treatment. Both the SIAS and Social Phobia Scale correlate strongly with each other [40–43], but while the latter was developed to assess fears of being observed or scrutinised by others, the SIAS was developed more specifically to assess fears and anxiety related to social interactions with others (e.g., meeting with others, initiating and maintaining conversations). The SIAS discriminates between clinical and non-clinical populations [40, 44, 45] and has also been found to differentiate between those with social anxiety and those with general anxiety [46], making it a useful clinical screening tool. Although originally developed in Australia, it has been tested and found to work well in diverse cultures worldwide [47–50], and has strong psychometric properties in clinical and non-clinical samples [40, 42, 43, 45–47].

For the current study, all 20 items of the SIAS were included in the survey, though we omitted the three positively-worded items from analyses, as studies have demonstrated that including them results in weaker than expected relationships between the SIAS and other measures, that they hamper the psychometric properties of the measure, and that the SIAS performs better without them [e.g., 51–53] (the omitted items were *'I find it easy to make friends my own age'*, *'I am at ease meeting people at parties, etc'*, and *'I find it easy to think of things to talk about'*.). One item of the SIAS was also modified prior to use: '*I have difficulty talking to attractive persons of the opposite sex*' was altered to '*I have difficulty talking to people I am attracted to*', to make it more applicable to individuals who do not identify as heterosexual, given that the original item was meant to measure difficulty talking to an attractive potential partner [54].

The questionnaire also included measures of resilience, in addition to other questions concerning functioning in daily life. These were included as part of a corporate social responsibility strategy to investigate the rates of social anxiety and resilience in each target market. A translation agency (Language Connect) translated the full survey into the national languages of the participants.

## Analyses

We analysed social anxiety scores for the overall sample, as well as by country, sex, and age (for sex, given the limited number and heterogeneity of individuals grouped into the 'other' category, we only compared males and females). As social anxiety is linked to work status [25], we also examined differences in SIAS scores between those working and those who were unemployed. Urban/rural differences were also investigated as previous research has suggested anxiety disorders may differ depending on where an individual lives [28]. Education level [27], too, was included using completion of secondary education (ISCED level 3) in a subgroup of participants aged 20 years and above to ensure all were above mandatory ages for completing high school. Descriptive statistics are reported for each group with significant differences explored using ANOVA (with Tukey post-hoc tests) or t-tests.

The SIAS is said to be unidimensional when using just the 17 straightforwardly-worded items [52], with item scores summed to give general social anxiety scores. Higher scores indicate greater levels of social anxiety. Heimberg and colleagues [42] have suggested a cut-off of 34 on the 20-item SIAS to denote a clinical level of social anxiety (SAD). This level has been adopted in other studies [e.g., 45] and found to accurately discriminate between clinical and non-clinical participants [53]. This threshold for SAD scales to 28.9 when just the 17 items are used, and this is slightly more conservative than others who have used 28 as an adjusted 17-item threshold [53, 55]. Therefore, in addition to analyses of raw scores to gauge the severity of social anxiety (and reflect consideration of social anxiety as a spectrum), we also report the proportion of individuals meeting or exceeding this threshold for SAD ($\geq$29) and analyse differences between groups using chi-square tests.

Additionally, despite the unidimensionality of the SIAS, the individual items can be interpreted as examples of contexts where social anxiety may be more or less acutely experienced (e.g., social situations with authority: '*I get nervous if I have to speak with someone in authority*', social situations with strangers: '*I am nervous mixing with people I don't know well*'). Therefore, as social anxiety may be experienced differently depending on culture [6], we also sorted the items in the measure to understand the top and least concerning contexts for each country.

Finally, we also sought to understand whether individuals perceived themselves as having social anxiety. After completing the SIAS, participants were presented with a definition of social anxiety and asked to reflect on whether they thought this was what they experienced. We contrasted responses with a SIAS threshold analysis to determine discrepancies, including assessment of the proportion of false positives (those who thought they had social anxiety but did not exceed the threshold) and false negatives (those who thought they did not have social anxiety but exceeded the threshold).

All analyses were conducted using SPSS v25 [56].

## Results

As the survey required a response for each item, there were no missing data. The internal reliability of the SIAS was found to be strong ($\alpha$ = .94), with the removal of any item resulting in a reduction in consistency.

### Social anxiety by sex, age, and country

In the overall sample, the distribution of social anxiety scores formed an approximately normal distribution with a slightly positive skew, indicating that most respondents scored lower than the midpoint on the measure (Fig 1). However, more than one in three (36%) were found to score above the threshold for SAD. There were no significant differences in social anxiety scores between male and female participants ($t(6768)$ = -1.37, n.s.) and the proportion of males and females scoring above the SAD threshold did not significantly differ either ($\chi^2(1,6770)$ = .54, n.s.).

Social anxiety scores significantly differed between countries ($F(6,6818)$ = 74.85, $p < .001$, $\eta_p^2$ = .062). Indonesia had the lowest average scores ($M$ = 18.94, $SD$ = 13.21) and the US had the highest ($M$ = 30.35, $SD$ = 15.44). Post-hoc tests revealed significant differences ($ps\leq.001$) between each of the countries, except between Brazil and Thailand, between China and Vietnam, between Russia and China, and between Russia and Indonesia (see Table 2). The proportion of individuals exceeding the threshold for SAD was also found to significantly differ between the seven countries ($\chi^2(6,6825)$ = 347.57, $p < .001$). Like symptom severity, the US had the highest prevalence with more than half of participants surveyed exceeding the threshold (57.6%), while Indonesia had the lowest, with fewer than one in four (22.9%).

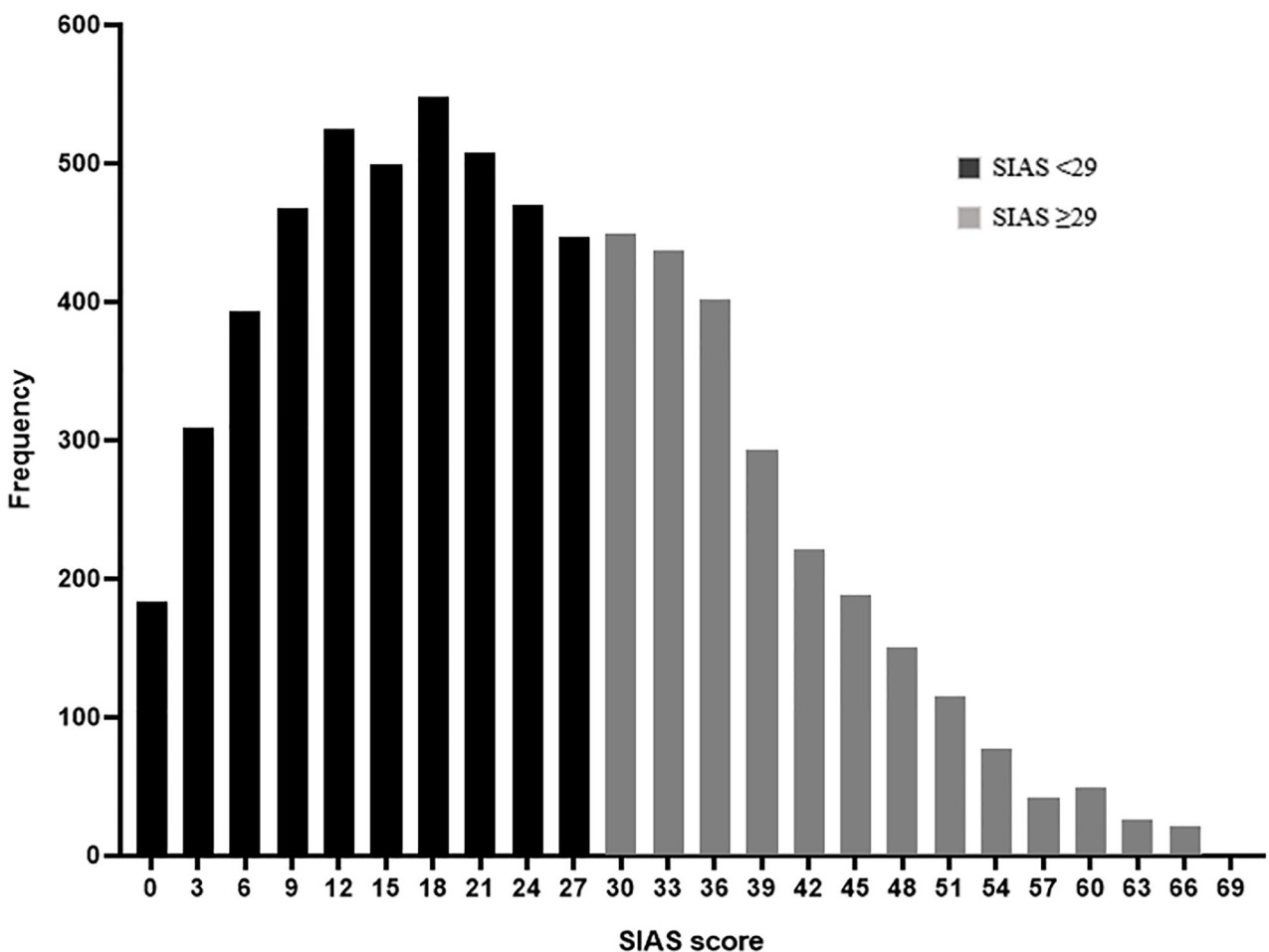

**Fig 1. Frequency of social anxiety scores (full sample).**

A significant age difference was also observed ($F(2,6822) = 39.74$, $p < .001$, $\eta_p^2 = .012$), where 18-24-year-olds scored significantly higher ($M = 25.33$, $SD = 13.98$) than both 16-17-year-olds ($M = 21.92$, $SD = 14.24$) and 25-29-year-olds ($M = 22.44$, $SD = 14.22$). Also, 25-29-year-olds scored significantly higher than 18-24-year-olds ($ps < .001$). The proportion of individuals scoring above the threshold for SAD also significantly differed between age groups ($\chi^2(2,6825) = 48.62$, $p < .001$) (Fig 2).

A three-way ANOVA confirmed significant main effect differences in social anxiety scores between age groups ($F(2,6728) = 38.93$, $p < .001$, $\eta_p^2 = .011$) and countries ($F(6,6728) = 45.37$, $p < .001$, $\eta_p^2 = .039$), as well as the non-significant difference between males and females ($F(1,6728) = .493$, n.s.). However, of the interactions between sex, age, and country, the two-way country*age interaction was significant ($F(12,6728) = 1.89$, $p = .031$, $\eta_p^2 = .003$), where 16-17-year-olds in Indonesia were found to have the lowest scores ($M = 15.70$, $SD = 13.46$) and 25-29-year-olds in the US had the highest ($M = 30.47$, $SD = 16.17$) (Fig 3). There was also a significant country*sex interaction ($F(6,6728) = 2.25$, $p = .036$, $\eta_p^2 = .002$), where female participants in Indonesia had the lowest scores ($M = 18.07$, $SD = 13.18$) and female participants in the US had the highest ($M = 30.37$, $SD = 15.11$) (Fig 4).

**Table 2. Social anxiety scores.**

| | SCORES | | SCORE DIFFERENCE BETWEEN GROUPS ($T$ / $F$, $P$) | PROPORTION WITH SAD (SIAS$\geq$29) (%) | PROPORTION DIFFERENCE BETWEEN GROUPS ($X^2$, $P$) |
|---|---|---|---|---|---|
| | *M* | *SD* | | | |
| Overall sample | 23.82 | 14.18 | | 36.2 | |
| *Sex* | | | -1.37, n.s. | | .54, n.s. |
| Male | 23.53 | 14.12 | | 35.6 | |
| Female | 24.00 | 14.18 | | 36.5 | |
| *Country* | | | 74.85, < .001 | | 347.57, < .001 |
| Brazil | 26.18 | 15.23 | | 42.4 | |
| China | 22.30 | 13.52 | | 32.1 | |
| Indonesia | 18.94 | 13.21 | | 22.9 | |
| Russia | 20.78 | 12.79 | | 27.0 | |
| Thailand | 25.57 | 13.92 | | 41.4 | |
| US | 30.35 | 15.44 | | 57.6 | |
| Vietnam | 22.68 | 11.77 | | 30.7 | |
| *Age* | | | 39.74, < .001 | | 48.62, < .001 |
| 16–17 | 21.92 | 14.24 | | 30.8 | |
| 18–24 | 25.33 | 13.98 | | 40.3 | |
| 25–29 | 22.44 | 14.22 | | 32.8 | |
| *Work* | | | 9.48, < .001 | | 7.55, .023 |
| Employed | 23.28 | 14.32 | | 35.3 | |
| Studying | 23.96 | 13.50 | | 36.5 | |
| Unemployed | 26.27 | 14.54 | | 41.7 | |
| *Urban/rural* | | | 9.95, < .001 | | 35.84, < .001 |
| Central urban | 22.70 | 14.67 | | 33.0 | |
| Urban area | 23.62 | 13.77 | | 35.3 | |
| Suburban | 25.64 | 14.08 | | 42.4 | |
| Semi-rural | 24.53 | 13.74 | | 37.9 | |
| Rural | 25.37 | 13.91 | | 41.9 | |
| *Education* | | | 5.51, < .001 | | 38.75, < .001 |
| L3 unfinished | 27.94 | 15.07 | | 52.0 | |
| L3 finished | 23.40 | 14.15 | | 34.8 | |

*M* = mean, *SD* = standard deviation, *t* = t-test, *F* = ANOVA, $\chi^2$ = chi-square, *p* = significance, L3 = ISCED level 3 (secondary education), SAD = Social Anxiety Disorder.

## Work status

Social anxiety scores were also found to significantly differ in terms of work status (employed/ studying/unemployed; $F(2,6030) = 9.48$, $p < .001$, $\eta_p^2 = .003$), with those in employment having the lowest scores ($M = 23.28$, $SD = 14.32$), followed by individuals who were studying ($M = 23.96$, $SD = 13.50$). Those who were unemployed had the highest scores ($M = 26.27$, $SD = 14.54$). Post-hoc tests indicated there were significant differences between those who were employed and unemployed ($p < .001$), between those studying and unemployed ($p = .006$), but not between those employed and those who were studying. The difference between those exceeding the SAD threshold between groups was also significant ($\chi^2(2,6033) = 7.55$, $p = .023$).

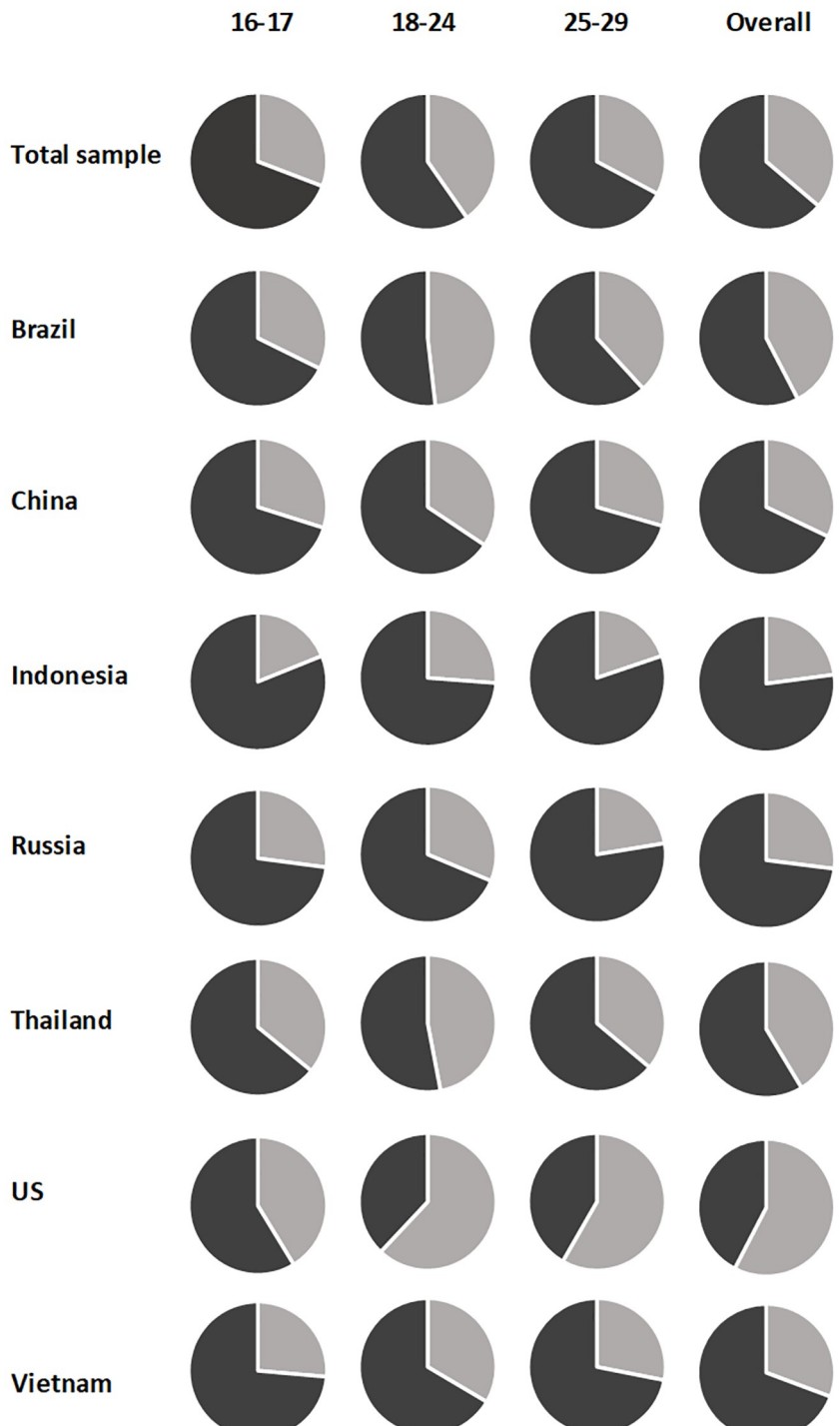

**Fig 2. Proportion of individuals meeting the threshold for Social Anxiety Disorder by age group and country.**

## Urban/Rural

Social anxiety scores also significantly varied depending on an individual's place of residence ($F(4,6820) = 9.95$, $p < .001$, $\eta_p^2 = .006$). However, this was not a linear relationship from urban

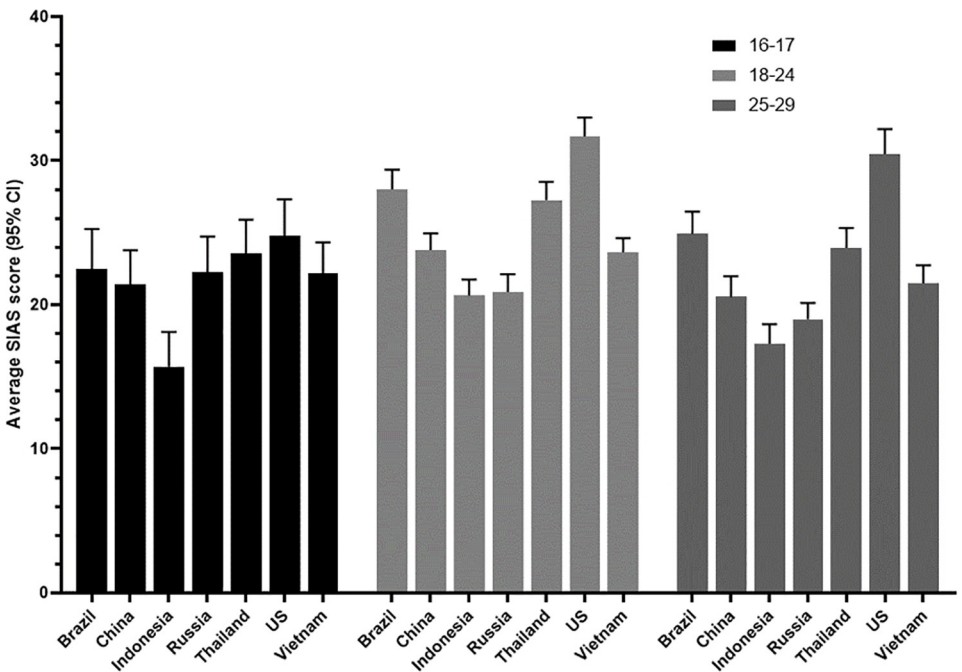

**Fig 3. Levels of social anxiety by country and age.**

to rural extremes (Fig 5); instead, those living in suburban areas had the highest scores ($M$ = 25.64, $SD$ = 14.08) and those in central urban areas had the lowest ($M$ = 22.70, $SD$ = 14.67). This pattern was reflected in the proportions of individuals exceeding the SAD threshold ($\chi^2(4,6825)$ = 35.84, $p < .001$).

## Education level

In the subsample of individuals aged 20 or above, level of education also resulted in a significant differences in social anxiety scores ($t(5071)$ = 5.51, $p < .001$), with individuals who completed secondary education presenting lower scores ($M$ = 23.40, $SD$ = 14.15) than those who had not completed secondary education ($M$ = 27.94, $SD$ = 15.07). Those exceeding the threshold for SAD also significantly differed ($\chi^2(1,5073)$ = 38.75, $p < .001$), with half of those who had not finished secondary education exceeding the cut-off (52%), compared to just over a third of those who had (35%).

## Concerns by context

Table 3 illustrates the items of the SIAS sorted by severity for each country. For East-Asian countries, speaking with someone in authority was a top concern, but less so for Brazil, Russia, and the US. Patterns became less discernible between countries beyond this top concern, indicating heterogeneity in the specific situations related to social anxiety, although individuals in most countries appeared to be least challenged by mixing with co-workers and chance encounters with acquaintances.

## Self-perceptions of social anxiety

Just over a third of the sample perceived themselves to experience social anxiety (34%). Although this was similar to the proportion of individuals who exceeded the threshold for

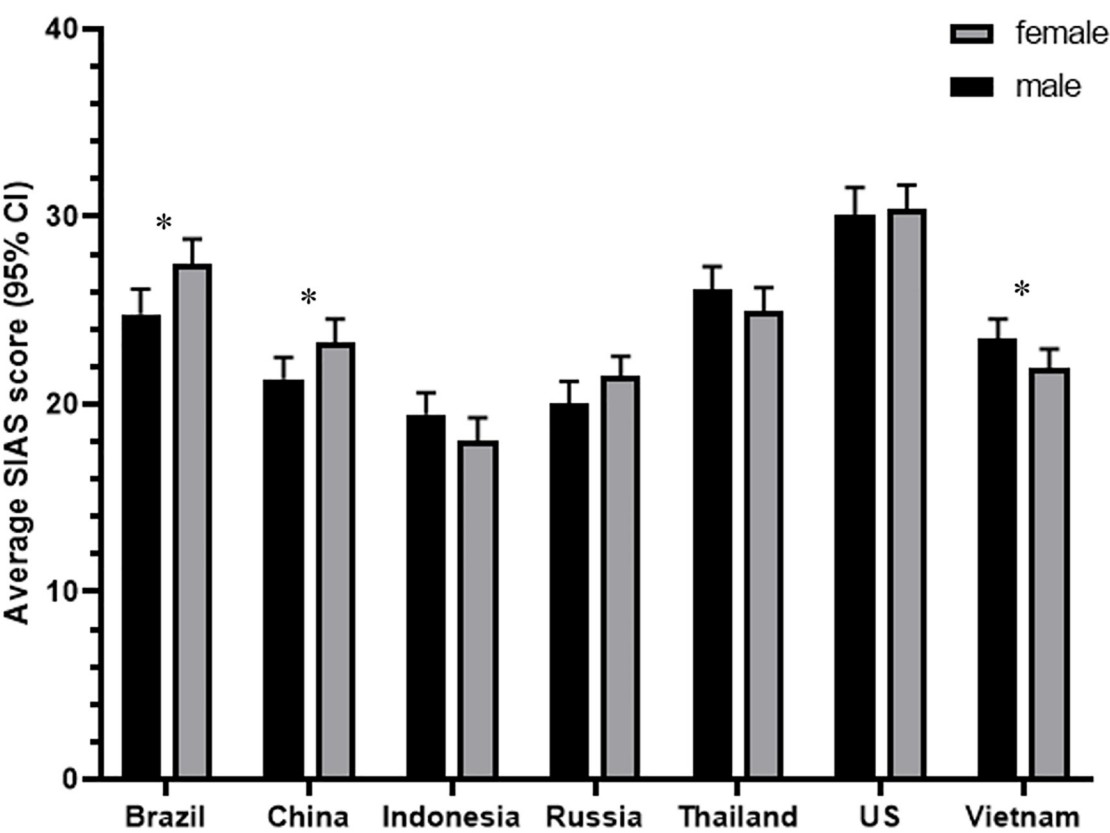

**Fig 4. Levels of social anxiety by country and sex.**

SAD (36%), perceptions significantly differed from threshold results ($\chi^2(1,6825) = 468.80$, $p <$ .001). Just fewer than half of the sample (48%) perceived themselves as not being socially anxious and were also below the threshold, and a fifth (18%) perceived themselves as being socially anxious and exceeded the threshold (Fig 6). However, 16% perceived themselves to be socially anxious yet did not exceed the threshold (false positives) and 18% perceived themselves not to be socially anxious yet exceeded the threshold (false negatives). This suggests a large proportion of individuals do not properly recognise their level of social anxiety (over a third of the sample), and perhaps most importantly, that more than 1 in 6 may experience SAD yet not recognise it (Table 4).

## Discussion

This study provides an estimate of the prevalence of social anxiety among young people from seven countries around the world. We found that levels of social anxiety were significantly higher than those previously reported, including studies using the 17-item version of the SIAS [e.g., 55, 57, 58]. Furthermore, our findings show that over a third of participants met the threshold for SAD (23–58% across the different countries). This far exceeds the highest of figures previously reported, such as Kessler and colleague's [16] lifetime prevalence rate of 12% in the US.

As this study specifically focuses on social anxiety in young people, it may be that the inclusion of older participants in other studies leads to lower average levels of social anxiety [27, 59]. In contrast, our findings show significantly higher rates of SAD than anticipated, and

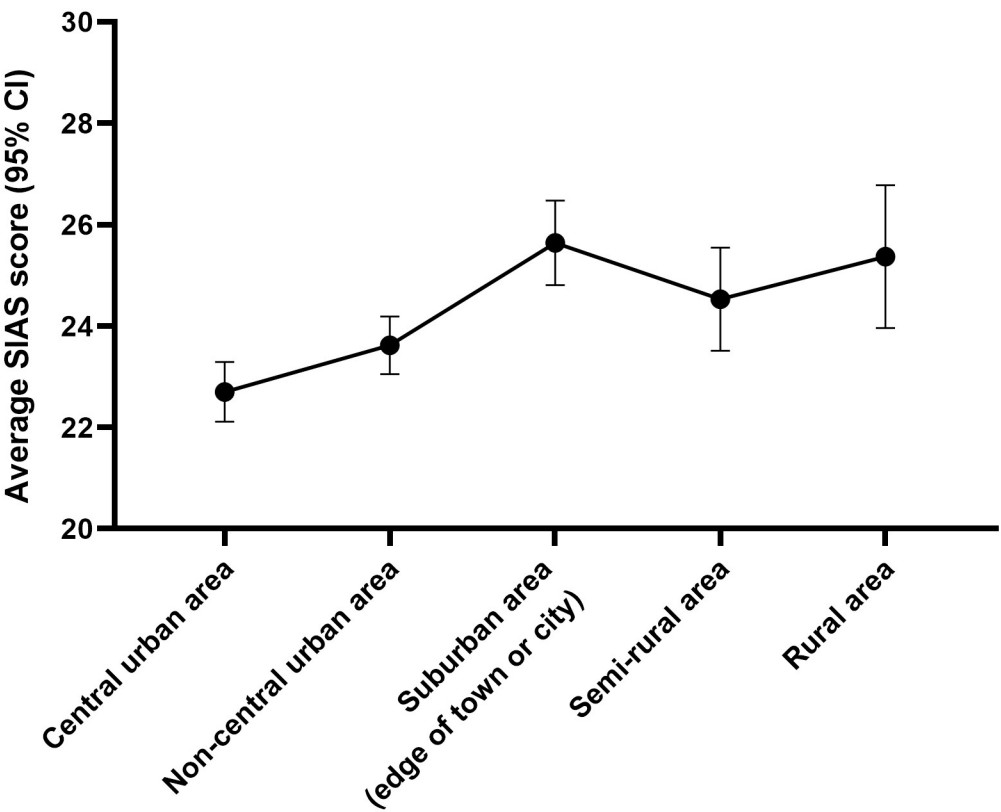

**Fig 5. Level of social anxiety by place of living.**

**Table 3. Concerns by country.**

| Item | | BR | CN | ID | RU | TH | US | VN | Overall |
|---|---|---|---|---|---|---|---|---|---|
| 1 | I get nervous if I have to speak with someone in authority (teacher, boss, etc.) | 5 | 1 | 1 | 4 | 1 | 5 | 3 | 1 |
| 2 | I have difficulty making eye contact with others | 11 | 11 | 10 | 9 | 14 | 12 | 12 | 12 |
| 3 | I become tense if I have to talk about myself or my feelings | 1 | 8 | 5 | 1 | 8 | 2 | 5 | 4 |
| 4 | I find it difficult to mix comfortably with the people I work with | 16 | 17 | 15 | 17 | 13 | 16 | 16 | 16 |
| 5 | I tense up if I meet an acquaintance in the street | 17 | 15 | 13 | 16 | 17 | 15 | 17 | 17 |
| 6 | When mixing socially, I am uncomfortable | 13 | 10 | 16 | 12 | 10 | 8 | 15 | 14 |
| 7 | I feel tense if I am alone with just one other person | 12 | 9 | 9 | 15 | 5 | 14 | 9 | 11 |
| 8 | I have difficulty talking with other people | 14 | 16 | 17 | 14 | 16 | 13 | 14 | 15 |
| 9 | I worry about expressing myself in case I appear awkward | 6 | 4 | 2 | 2 | 6 | 3 | 1 | 3 |
| 10 | I find it difficult to disagree with another's point of view | 15 | 12 | 11 | 13 | 7 | 17 | 6 | 13 |
| 11 | I have difficulty talking to people I am attracted to | 3 | 13 | 4 | 8 | 4 | 6 | 8 | 7 |
| 12 | I find myself worrying that I won't know what to say in social situations | 4 | 3 | 8 | 7 | 3 | 4 | 4 | 5 |
| 13 | I am nervous mixing with people I don't know well | 2 | 5 | 3 | 6 | 2 | 1 | 2 | 2 |
| 14 | I feel I'll say something embarrassing when talking | 7 | 14 | 14 | 3 | 15 | 7 | 7 | 10 |
| 15 | When mixing in a group, I find myself worrying I will be ignored | 8 | 7 | 7 | 10 | 11 | 11 | 11 | 8 |
| 16 | I am tense mixing in a group | 10 | 6 | 12 | 11 | 12 | 9 | 13 | 9 |
| 17 | I am unsure whether to greet someone I know only slightly | 9 | 2 | 6 | 5 | 9 | 10 | 10 | 6 |

Dark shaded cells indicate the top three concerns (1–3); lightly shaded cells indicate the least three concerns (15–17); BR = Brazil; CN = China; ID = Indonesia; RU = Russia; TH = Thailand; US = United States; VN = Vietnam.

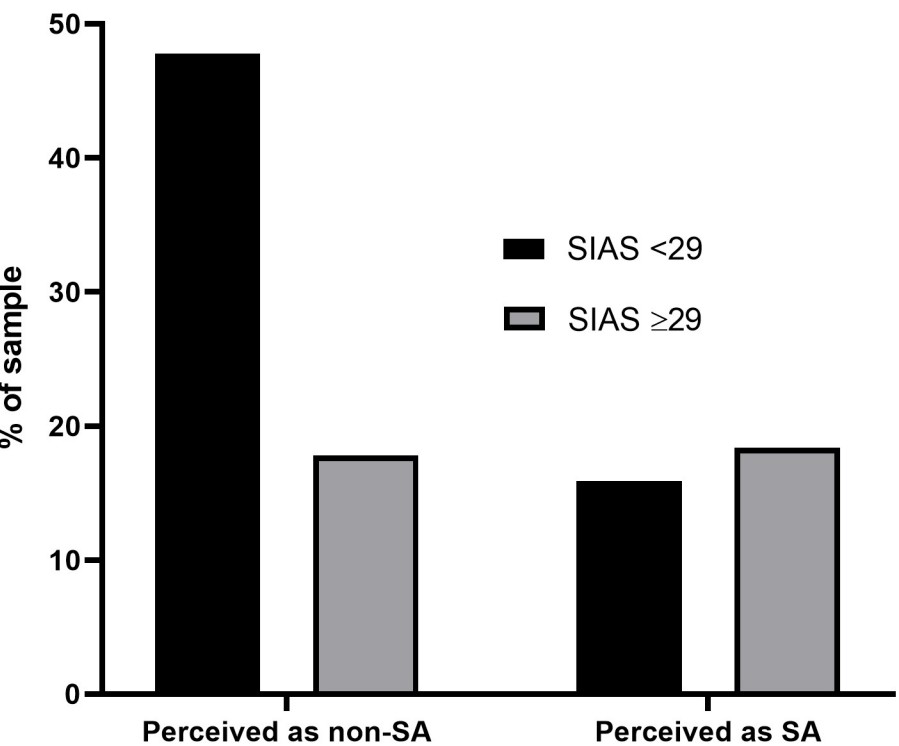

**Fig 6. Perceptions of social anxiety vs. classification.**

particularly so for individuals aged 18–24. It also extends the argument of authors such as Lecrubier and colleagues [60] and Leigh and Clark [30] that developmental challenges during adolescence may provoke social anxiety, especially the crucial later period when leaving school and becoming more independent.

**Table 4. Classification of social anxiety scores.**

| | Self-perceived as non-SA | | | | | | Self-perceived as SA | | | | | |
|---|---|---|---|---|---|---|---|---|---|---|---|---|
| | SIAS <28 [a] | | | SAD: SIAS ≥28 [b] | | | SIAS <28 [b] | | | SAD: SIAS ≥28 [a] | | |
| | *M* | *SD* | *% of group* | *M* | *SD* | *% of group* | *M* | *SD* | *% of group* | *M* | *SD* | *% of group* |
| Overall sample | 13.92 | 7.80 | 47.8 | 38.53 | 8.38 | 17.8 | 18.34 | 6.79 | 15.9 | 40.02 | 8.60 | 18.4 |
| *Sex* | | | | | | | | | | | | |
| Male | 13.96 | 7.82 | 50.1 | 37.56 | 8.42 | 20.5 | 18.26 | 7.02 | 14.3 | 39.87 | 8.39 | 15.1 |
| Female | 13.92 | 7.77 | 46.0 | 38.42 | 8.29 | 15.1 | 18.42 | 6.60 | 17.6 | 40.09 | 8.73 | 21.4 |
| *Country* | | | | | | | | | | | | |
| Brazil | 14.10 | 8.15 | 39.4 | 39.36 | 8.37 | 16.2 | 17.88 | 6.99 | 18.2 | 41.96 | 9.19 | 26.2 |
| China | 13.18 | 7.60 | 46.4 | 38.73 | 8.08 | 15.4 | 17.78 | 6.40 | 21.6 | 38.36 | 7.56 | 16.6 |
| Indonesia | 12.49 | 8.03 | 65.0 | 37.42 | 8.45 | 15.3 | 17.62 | 7.32 | 12.1 | 38.89 | 8.47 | 7.6 |
| Russia | 13.51 | 7.61 | 57.1 | 36.09 | 6.28 | 13.9 | 18.24 | 6.49 | 16.0 | 39.37 | 7.64 | 13.0 |
| Thailand | 15.15 | 7.60 | 46.7 | 38.76 | 8.79 | 21.8 | 19.48 | 6.26 | 11.9 | 39.38 | 8.64 | 19.6 |
| US | 13.79 | 8.70 | 27.4 | 40.95 | 9.32 | 25.7 | 19.09 | 6.70 | 14.8 | 41.35 | 9.15 | 31.5 |
| Vietnam | 15.58 | 6.82 | 52.7 | 36.55 | 6.88 | 16.4 | 18.74 | 6.49 | 16.6 | 37.57 | 7.01 | 14.3 |

*M* = Mean, *SD* = Standard Deviation.

[a] Congruence: self-perceptions align with measure.

[b] Conflict in classification (false positive or negative).

We also found strong variations in levels of social anxiety between countries. Previous explorations of national prevalence rates have been less equivocal, with some reporting differences [6] while others have not [61]. Our findings concur with those of Hofmann and colleagues' [6] who note that the US has typically high rates of social anxiety, which we also found (in contrast to other countries). However, the authors suggest Russia also has a high prevalence and that Asian cultures typically show lower rates. In contrast, we found samples from Asian countries such as Thailand and Vietnam had higher rates than in the sample from Russia, and that there were significant differences between Asian countries themselves (Table 2). As our study used the SIAS, which determines how socially anxious an individual is based on their ratings of difficulty in specific social situation, one way of accounting for differences may be to consider the kinds of feared social situations that are covered in the measure. For instance, our breakdown of concerns by country (Table 3) indicates that in Asian countries, speaking with individuals in authority is a strongly feared situation, but this is less challenging in other cultures. For non-Asian countries, one of the strongest concerns was talking about oneself or one's feelings. In Asian countries, where there is typically less of an emphasis on individualism, talking about oneself may be less stressful if there is less perceived pressure to demonstrate one's uniqueness or importance. Future investigations could further explore cultural differences in social anxiety across different types of social situations or could confirm cross-cultural social anxiety heterogeneity by using approaches that are less heavily tied to determining social anxiety within given contexts (e.g., a diagnostic interview), as many of the commonly used measures appear to be [62, 63].

Our findings also provide mixed support for investigations of other demographic differences in social anxiety. First, previous studies have tended to indicate that female participants score higher than males on measures of social anxiety [27, 64]. Although the samples from Brazil and China reflected this, we found no difference between males and females in the overall sample, nor in samples from Indonesia, Russia, Thailand, US, or Vietnam. Sex-related differences in social anxiety have been attributed to gender differences, such as suggestions that girls ruminate more, particularly about relationships with others [65, 66]. It is possible that as gender roles and norms vary between countries, and in some instances start to decline, so may differences in social anxiety, which younger generations are likely to reflect first. However, given the unexpected finding that males in Vietnam scored significantly higher than their female counterparts, further investigation is needed to account for the potentially culturally nuanced relationship between sex and social anxiety.

We also confirmed previous findings that higher levels of social anxiety are associated with lower levels of education and being unemployed. Although these findings are in-line with previous research [27, 64], our study cannot shed light on causal mechanisms; longitudinal research is required to establish whether social anxiety leads individuals to struggle with school and work, whether struggling in these areas provokes social anxiety, or whether there is a more dynamic relationship.

Finally, we found that 18% of the sample could be classified as "false negatives". This sizeable group felt they did not have social anxiety, yet their scores on the SIAS considerably exceeded the threshold for SAD. It has been said that SAD often remains undiagnosed [67], that individuals who seek treatment only do so after 15–20 years of symptoms [68], and that SAD is often identified when a related condition warrants attention (e.g., depression or alcohol abuse; Schneier [5]). It has also been reported that many individuals do not recognise social anxiety as a disorder and believe it is just part of their personality and cannot be changed [3]. Living with an undiagnosed or untreated condition can result in substantial economic consequences for both individuals and society, including a reduced ability to work and a loss of productivity [69], which may have a greater impact over time compared to those who receive

successful treatment. Furthermore, the variety of avoidant (or "safety") behaviours commonly associated with social anxiety [70, 71] mean that affected individuals may struggle or be less able to function socially, and for young people at a time in their lives when relationships with others are particularly crucial [72, 73], the consequences may be significant and lasting. Greater awareness of social anxiety and its impact across different domains of functioning may help more young people to recognise the difficulties they experience. This should be accompanied by developing and raising awareness of appropriate services and supports that young people feel comfortable using during these important developmental stages [see 30, 74].

## Study limitations

Our ability to infer reasons for the prevalence of SAD is hindered by the present data being cross-sectional, and therefore only allowing for associations to be drawn. We are also unable to confirm the number of clinical cases in the sample, as we did not screen for those who may have received a professional diagnosis of SAD, nor those who are receiving treatment for SAD. Additionally, the use of an online survey incorporating self-report measures incurs the risk of inaccurate responses. Further research could build on this investigation by surveying those in middle and older age to discover whether rates of social anxiety have also risen across other ages, or whether this increase is a youth-related phenomenon. Future investigations could also use diagnostic interviews and track individuals over time to determine the onset and progression of symptoms, including whether those who are subclinical later reach clinical levels, or vice versa, and what might account for such change.

## Conclusion

On a global level, we report higher rates of social anxiety symptoms and the prevalence of those meeting the threshold for SAD than have been reported previously. Our findings suggest that levels of social anxiety may be rising among young people, and that those aged 18–24 may be most at risk. Public health initiatives are needed to raise awareness of social anxiety, the challenges associated with it, and the means to combat it.

## Acknowledgments

The authors would like to acknowledge the role of Edelman Intelligence for collecting the original data on behalf of Unilever and CLEAR as part of their mission to support the resilience of young people.

## Author Contributions

**Conceptualization:** Philip Jefferies, Michael Ungar.

**Formal analysis:** Philip Jefferies.

**Methodology:** Philip Jefferies, Michael Ungar.

**Writing – original draft:** Philip Jefferies.

**Writing – review & editing:** Philip Jefferies, Michael Ungar.

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
