## [Decision Letter · Decision Letter 0]

1 Jul 2020

PONE-D-20-06197

Social anxiety in young people: A prevalence study in seven countries

PLOS ONE

Dear Dr. Jefferies,

Thank you for submitting your manuscript to PLOS ONE. After careful consideration, we feel that it has merit but does not fully meet PLOS ONE’s publication criteria as it currently stands. Therefore, we invite you to submit a revised version of the manuscript that addresses the points raised during the review process.

We look forward to receiving your revised manuscript.

Kind regards,

Sarah Hope Lincoln

Academic Editor

PLOS ONE

Additional Editor Comments:

Thank you for your submission. The reviewers felt like this paper is important for the field, but they have some reservations about various aspects of the manuscript. We invite you to revise and resubmit.

'The authors would like to acknowledge the role of Edelman Intelligence for collecting the data and Unilever and CLEAR for funding and commissioning the overarching project as part of their mission to support the resilience of young people experiencing social anxiety.'

'The author(s) received no specific funding for this work.'

Additionally, because some of your funding information pertains to commercial funding, we ask you to provide an updated Competing Interests statement, declaring all sources of commercial funding.

In your Competing Interests statement, please confirm that your commercial funding does not alter your adherence to PLOS ONE Editorial policies and criteria by including the following statement: "This does not alter our adherence to PLOS ONE policies on sharing data and materials.” as detailed online in our guide for authors  http://journals.plos.org/plosone/s/competing-interests.  If this statement is not true and your adherence to PLOS policies on sharing data and materials is altered, please explain how.

Please include the updated Competing Interests Statement and Funding Statement in your cover letter. We will change the online submission form on your behalf.

4. Please amend your manuscript to include your abstract after the title page.

Reviewers' comments:

Reviewer's Responses to Questions

**Comments to the Author**

1. Is the manuscript technically sound, and do the data support the conclusions?

Reviewer #1: Yes

Reviewer #2: Yes

2. Has the statistical analysis been performed appropriately and rigorously? 

Reviewer #1: Yes

Reviewer #2: Yes

3. Have the authors made all data underlying the findings in their manuscript fully available?

Reviewer #1: Yes

Reviewer #2: Yes

4. Is the manuscript presented in an intelligible fashion and written in standard English?

Reviewer #1: Yes

Reviewer #2: Yes

5. Review Comments to the Author

Reviewer #1: This study provides valuable information for both researchers and clinicians specializing in anxiety as it offers an examination of cross-cultural and worldly prevalence rates of SAD. In addition to examining prevalence rates between seven countries, the authors examine SAD rates through contributing factors such as gender, age, and SES. The strengths of this paper include the evolutionary perspective of social anxiety and making the distinction between subclinical social anxiety and symptoms that reach the clinical threshold of SAD. The authors also present multiple graphs of their findings that are visually appealing and easy to understand. There are a number of suggested changes that would strengthen this manuscript, which are presented below.

Major:

1. The authors can improve the quality of this manuscript by expanding on many ideas and theories that are only briefly mentioned. This can be accomplished both by incorporating more background literature as well as the authors’ own interpretations and hypotheses. These include:

Introduction:

-Speak more to the distinction between subthreshold social anxiety and clinical levels of SAD. For example, are there any predisposing factors for some to reach clinical levels of social anxiety compared to others? Are there any current theories (potentially physiological in nature) that explain the differences between subclinical and clinical social anxiety?

-When discussing current prevalence rates of SAD, the authors should consider also including rates of undiagnosed SAD and implications on treatment. The authors first include this information in the discussion starting on line 331 but it should be first presented in the introduction.

-It may be helpful to include a developmental theory of SAD in the introduction, especially as the authors repeatedly reference social media and technology contributing to the development of SAD in young adults.

-Likewise, given the focus on worldly prevalence of SAD, further detail on cross-cultural perspectives is likely merited. The authors briefly social anxiety in Asian cultures but should offer more information regarding other cultures. Additionally, the authors could provide hypotheses from the existing literature as to why there are differing rates of social anxiety in different parts of the world rather than simply stating the statistics in the introduction (i.e., the differences in social anxiety expression that the authors note in the discussion). For example, the authors find that speaking with authority figures to be a top concern for Asian countries compared to other countries, which may potentially be linked with cultural values of respect to authority and status that could discussed in the introduction.

-It would be helpful for readers to have some explanation (either hypothesized by the existing literature or the authors’ own speculation) as to why social anxiety rates are increased by social media/technology and why this rate is hypothesized to increase for only young adults. This may be achieved through the Lecrubier et al. and Leigh and Clark’s theories that are referenced in the discussion.

-The authors introduce contributing factors of status, location, and education in the findings but do not discuss these factors in the introduction (i.e., line 315-316 females scoring higher on SAD measures than males). Consider presenting these factors and their associated literature earlier on in the manuscript given they are major areas of the study’s focus.

-The authors do not provide a reason or explanation for their study. Consider clarifying the importance of this particular study and reasoning behind specific aims (Why young adults with SAD? Why assess cross-cultural prevalence?). Was there any reasoning behind choosing the seven countries that were included in this study?

Discussion:

-Before line 299, it is suggested that the authors first reiterate and elaborate on their findings regarding age and social anxiety before their discussion of these particular findings. The authors may want to further elaborate on their findings of age and social anxiety. Do these findings predict an increase in the rates of SAD in the future? What are the clinical implications of this potential high-risk age range?

-Expand on findings related to cross-cultural differences in prevalence (paragraph starting line 307). Why might these discrepancies exist? What are the implications of this finding? Additionally, the two sentences starting at line 309 (Our findings concur…) ending at line 313 are confusing to the reader. Consider rephrasing or possibly listing countries with the highest to lowest rates of endorsement for social anxiety.

-Explain potential reasoning behind the differences seen in male/female prevalence rates (paragraph starting line 314). Are these findings in line with the existing literature on gender differences in prevalence rates in these respective countries? Why might these differences exist in this particular sample?

-The authors should consider including additional implications of incorrectly perceived social anxiety in the paragraph starting on 329.

-There are additional study limitations that the authors do not address in the manuscript. The most notable limitation is the use of an online survey to gather responses. The authors cannot be confident that participants are accurate in their reports of demographics and social anxiety symptoms. Additionally, the authors were only able to utilize one questionnaire of social anxiety rather a diagnostic interview and are unable to gather important information such as previous/current treatment or medication management of symptoms. Relatedly, the authors were unable to include additional measures beyond social anxiety symptoms that could have contributed to their findings, such as social media usage questionnaires.

-Consider including a section on the clinical implications of these findings or include these implications throughout the discussion.

Minor:

2. To improve the flow of the article, it is suggested that the authors move lines 32-34 (starting with “individuals experiencing social anxiety visibly struggle…” to after line 29 “boring or incompetent.” This allows for line 35 to continue with the authors’ discussion on impairment.

3. Consider adding race/ethnicity and SES to Table 1’s demographics.

4. On line 116, please provide which three items from the scales were not included in analyses.

5. The paragraph starting on line 314 continually indicates scores are “worse” or “poorly.” Consider rephrasing to clarify if authors mean score higher/lower.

Reviewer #2: This is a well-written manuscript addressing an important topic. Strengths include the large sample, and cross-national comparisons.

The most significant limitation of the current manuscript is not carefully distinguishing between heightened social anxiety vs social anxiety disorder. Relying on cutoff scores on the SIAS greatly overestimates the prevalence of social anxiety disorder. Given that the authors report that they collected data on functioning (see page 7), I would have liked to see the authors incorporate impairment in functioning in their determination of "disorder".

6. PLOS authors have the option to publish the peer review history of their article (what does this mean?). If published, this will include your full peer review and any attached files.

Reviewer #1: No

Reviewer #2: No

---

## [Author Response · Author response to Decision Letter 0]

14 Jul 2020

Please see the Response to Reviewers file for comments and details of specific changes made to address each issue.

---

## [Decision Letter · Decision Letter 1]

1 Sep 2020

Social anxiety in young people: A prevalence study in seven countries

PONE-D-20-06197R1

Dear Dr. Jefferies,

We’re pleased to inform you that your manuscript has been judged scientifically suitable for publication and will be formally accepted for publication once it meets all outstanding technical requirements.

Kind regards,

Sarah Hope Lincoln

Academic Editor

PLOS ONE

Additional Editor Comments (optional):

Reviewers' comments:

Reviewer's Responses to Questions

**Comments to the Author**

1. If the authors have adequately addressed your comments raised in a previous round of review and you feel that this manuscript is now acceptable for publication, you may indicate that here to bypass the “Comments to the Author” section, enter your conflict of interest statement in the “Confidential to Editor” section, and submit your "Accept" recommendation.

Reviewer #1: All comments have been addressed

2. Is the manuscript technically sound, and do the data support the conclusions?

Reviewer #1: (No Response)

3. Has the statistical analysis been performed appropriately and rigorously? 

Reviewer #1: (No Response)

4. Have the authors made all data underlying the findings in their manuscript fully available?

Reviewer #1: (No Response)

5. Is the manuscript presented in an intelligible fashion and written in standard English?

Reviewer #1: (No Response)

6. Review Comments to the Author

Reviewer #1: (No Response)

7. PLOS authors have the option to publish the peer review history of their article (what does this mean?). If published, this will include your full peer review and any attached files.

Reviewer #1: No

---

## [Editor Report · Acceptance letter]

3 Sep 2020

PONE-D-20-06197R1 

Social anxiety in young people: A prevalence study in seven countries 

Dear Dr. Jefferies:

I'm pleased to inform you that your manuscript has been deemed suitable for publication in PLOS ONE. Congratulations! Your manuscript is now with our production department. 

Kind regards, 

on behalf of

Dr. Sarah Hope Lincoln 

Academic Editor

PLOS ONE